# Spiral Chiral Metamaterial Structure Shape for Optical Activity Improvements

**DOI:** 10.3390/mi14061156

**Published:** 2023-05-30

**Authors:** Kohei Maruyama, Miyako Mizuna, Takuya Kosuge, Yuki Takeda, Eiji Iwase, Tetsuo Kan

**Affiliations:** 1School of Informatics and Engineering, The University of Electro-Communications, Tokyo 182-8585, Japan; maruyama@ms.mi.uec.ac.jp; 2Department of Applied Mechanics and Aerospace Engineering, Waseda University, Tokyo 169-8555, Japan; mizuna@iwaselab.amech.waseda.ac.jp (M.M.); takeda@iwaselab.amech.waseda.ac.jp (Y.T.); iwase@waseda.jp (E.I.); 3Department of Mechanical and Intelligent Systems Engineering, Graduate School of Informatics and Engineering, The University of Electro-Communications, Tokyo 182-8585, Japan; kosuge@ms.mi.uec.ac.jp; 4Kagami Memorial Research Institute for Materials Science and Technology, Waseda University, Tokyo 169-0051, Japan

**Keywords:** chiral metamaterial, spiral shape, laser writing, GHz, optical activity

## Abstract

We report on a spiral structure suitable for obtaining a large optical response. We constructed a structural mechanics model of the shape of the planar spiral structure when deformed and verified the effectiveness of the model. As a verification structure, we fabricated a large-scale spiral structure that operates in the GHz band by laser processing. Based on the GHz radio wave experiments, a more uniform deformation structure exhibited a higher cross-polarization component. This result suggests that uniform deformation structures can improve circular dichroism. Since large-scale devices enable speedy prototype verification, the obtained knowledge can be exported to miniaturized-scale devices, such as MEMS terahertz metamaterials.

## 1. Introduction

In recent years, research on optical metamaterials has been actively investigated as a technology to obtain arbitrary optical properties using artificial structures [1,2,3]. By arraying fine structures below the wavelength, it is possible to perform a wide range of operations on electromagnetic waves, such as wavelength filtering [4,5,6], phase modulation, and polarization control [7,8,9,10]. Chiral metamaterials are a type of optical metamaterial [11,12,13,14,15,16,17,18]. Chirality is a property in which the structure reflected in a mirror does not overlap with the original form and does not have a mirror symmetry plane, similarly to the right and left hands of a human. When electromagnetic waves pass through chiral materials, a difference in the refractive index is observed depending on whether the polarization state of the electromagnetic wave is right-circularly polarized or left-circularly polarized, resulting in a phenomenon known as optical rotation or circular birefringence [19]. In addition, for right-handed and left-handed structures or mirror-image enantiomers, the direction of rotation of the polarization plane is reversed due to circular birefringence, even if the transmitted electromagnetic wave intensity is the same. Due to these different transmission characteristics, chiral materials exhibit such significant optical properties for right- and left-circularly polarized waves. The phenomenon of showing different transmission properties due to right- and left-circular polarizations is called circular dichroism, but no objects that exhibit strong circular dichroism exist in nature [20]. However, it is possible to achieve strong circular dichroism by constructing metamaterial structures using chiral materials. Therefore, chiral metamaterials have been studied as a promising method for polarization control.

By constructing chiral materials with an artificial structure, it becomes possible to change their transmission characteristics in response to external controls, making it a very important technology for applications in measurement and control [21,22,23,24,25]. Particularly in wavelength bands with long wavelengths of several hundred micrometres, such as terahertz (THz), chiral metamaterials can be constructed using a microelectromechanical system (MEMS). By adding dynamic structural deformation to the chiral structure, significant polarization modulation capabilities can be obtained [18,26]. For example, a metamaterial composed of flat spiral structures with a diameter of approximately 150 µm is known to form a three-dimensional spiral structure by deforming out of the substrate plane with an external force, exhibiting a very large circular dichroism. Furthermore, by changing the out-of-plane deformation direction of the flat spiral structure, it is possible to change the right-handed and left-handed structures with the same structure. Thus, a significantly higher degree of freedom in polarization modulation can be achieved compared to conventional methods. However, at present, chiral metamaterials with such three-dimensional spiral structures can be further improved structurally; for example, they may be made to achieve large and broadband circular dichroism. According to computational analysis, it is expected that an increase in circular dichroism will be made possible by improving the deformation shape of the spiral (Figure 1). However, there is no investigation estimating the shape of the three-dimensional spiral structure after deformation and evaluating the improvement in the optical activity caused by the change in the spiral shapes.

The aim of this study is to improve the circular dichroism of spiral-shaped chiral metamaterials by examining the shape of the spiral structure after deformation. First, an approximate model of the deformed spiral structure is constructed based on material mechanics. A theoretical verification of how the conventional spiral structure is deformed is carried out. Then, experimental verification is performed to determine whether similar deformation can be achieved using a pulling method at the center, which allows for significant deformation. In this experiment, large-scale device verification in the GHz frequency was adopted to speed up the improvement of the device’s performance. In the verification step, we compared the deformation manner of the large-scale spiral with the micro-sized spirals and verified the similarity of the deformation regardless of the size scale. The three-dimensional deformation shape and resulting polarization characteristics were measured using GHz-band electromagnetic waves.

## 2. Materials and Methods

### 2.1. Numerical Analysis

To investigate the changes in optical activity depending on the spiral shape, electromagnetic field simulations using the finite element method (COMSOL Multiphysics, ver. 5.5, Burlington, MA, USA) were performed. The method of electromagnetic field simulation is described in detail in [18], so we omit the details here. As shown in Figure 1a, we calculated the electromagnetic wave response of an asymmetric spiral structure where the upward deformation is reduced as it goes towards the center of the spiral, which corresponds to the experimentally observed results, and a spiral structure where the deformation uniformly changes according to θ from the outer periphery to the inner spiral. The cells have a size of 170 µm in the depth direction and in the plane of the paper and are 600 µm in the vertical direction. The boundary condition of the cell was the Floquet periodic condition. Only the spiral part is made of metal, and the original planar spiral with a diameter of 150 µm at the outer periphery deforms under force in the direction outside the central plane (the upwards direction in the figure) to form a three-dimensional spiral. The base of the spiral is fixed, and the structure is assumed to have a deformation of 300 µm at both the base and the center. Left- and right-circularly polarized light was irradiated on this structure, and the ellipticity angle was derived from the transmittance using the graph in Figure 1b. The ellipticity angle is derived from the ratio of transmittance of left- and right-circularly polarized light and can be described by the following Equation (1):(1)$$\mathrm{Ellipticity}=\tan^{-1}\frac{T_{\mathrm{LCP}}-T_{\mathrm{RCP}}}{T_{\mathrm{LCP}}+T_{\mathrm{RCP}}}$$
where T_LCP_ and T_RCP_ indicate the transmittance of the left-handed and right-handed circularly polarized light, respectively. The maximum value of ellipticity is ±45°, which corresponds to a strong polarization ability that completely shuts out the transmittance of one of the circular polarizations and transmits only the other. A larger absolute value indicates a strong interaction with circular polarization. From the graph, it is suggested that, when a uniform deformation is applied, there is a slight increase in the maximum absolute value, and good characteristics, such as the broadband of strong circular dichroism, are observed. Therefore, we will discuss the material mechanics considerations regarding the deformation shape of this spiral structure, followed by its fabrication, based on these considerations.

### 2.2. Deformation Theory of the Spiral Structure

The spiral structure used in this study was designed to be an Archimedes’ spiral, as shown in Figure 2. The radius line of the spiral structure, depicted as a dotted line in Figure 2a, is expressed as Equation (2):(2)$$r\left(\theta \right)=r_{\max}\left(1-\frac{\theta }{m\pi }\right)$$
*r*(*θ*) is the distance from the origin at the azimuth angle *θ*, *r*_max_ is the maximus radius at *θ* = 0, and *mπ* corresponds to the maximus angle of the spiral; in this case, 10*π*. *n* is the number of turns, and the range of the azimuthal angle is 0≤θ≤2nπ. The spiral in this paper is a 3-turn spiral, with *n* = 3 and 0≤θ≤6π. We modelled the spiral such that the beam width changes depending on *θ*, defined as *w*(*θ*).

When a spiral structure is deformed into a three-dimensional shape by the load *W* in the vertical out-of-plane direction (Figure 2b), the deformation properties can be mostly explained by a pure vertical deformation *δ*(*θ*) and the vertical displacement at *θ* (Figure 2c). Pure perpendicular deformation is the deformation component where all points of the spiral structure are deformed perpendicular to the horizontal substrate when the center of the spiral structure is pulled. Regarding the lengths of the beams of a spiral structure, which is the total length along the centerline of the spiral *l*(*θ*) (Figure 2b), the microelement *dl* in the mediating variable representation is as follows:(3)$$dl=\sqrt{r^{2}\left(\theta \right)+{\left(\frac{dr\left(\theta \right)}{d\theta }\right)}^{2}}d\theta \;.$$
The pure vertical deformation *δ*(*θ*) is thus expressed by the following equation using the load *W*, the transverse modulus of elasticity *G*, the sectional quadratic polar moment *I_p_*, the radius of the planar spiral structure *r*(*θ*), and the spiral length *l*(*θ*):(4)$$\delta \left(\theta_{0}\right)=\int_{0}^{\theta_{0}}d\delta \left(\theta \right)=\int_{0}^{l}\frac{Wr{\left(\theta \right)}^{2}}{GI_{p}}dl=\frac{Wr_{max}{}^{3}}{GI_{p}}\int_{0}^{\theta_{0}}{\left(1-\frac{\theta }{m\pi }\right)}^{2}\sqrt{1+{\left(1-\frac{\theta }{m\pi }\right)}^{2}}d\theta \;\;.$$
Figure 3 shows the calculated out-of-plane deformation of the spiral structure with a constant *w*(*θ*). The vertical axis represents the amount of vertical out-of-plane deformation, which is normalized to a maximum of 1, and the horizontal axis represents the azimuthal angle *θ*. In Equation (4), the load *W*, the transverse modulus of elasticity *G*, the sectional quadratic polar moment *I_p_*, and the constant radius *r*_max_ of the outermost circumference of the plane spiral structure, which are related to the properties of materials and designs, do not depend on *θ*. The theory indicates that the normalized out-of-plane deformation distribution remains the same regardless of those parameters. The electromagnetic simulation in Figure 1 suggests that the optical activity improves if the vertical deformation *δ*(*θ*) is uniform with respect to the azimuth angle, as indicated by the dotted line in Figure 3. The outer circumference of the planar spiral is the root of the beam and is subject to large deformation. Therefore, to suppress the deformation at the root, it is useful to increase the beam width at the root, which leads to an increase in the sectional quadratic polar moment *I_p_*. If the sectional quadratic polar moment *I_p_* at the root of the beam is increased, it is expected that the deformation will approach uniformity from the outermost circumference to the center of the beam.

### 2.3. Design of the Spiral Structure

The material of the structure was an acrylic plate of 0.5 mm thickness, which was suitable for laser processing. The design parameters of the metamaterials are given below. The maximum radius of the spiral *r*_max_ was 17.05 mm, the number of turns was 3, and the azimuthal angle *θ* ranged from 0 rad to 6π rad. The azimuthal angle was 0 rad at the root of the outer circumference and increased counterclockwise in the inner circumference direction. The beam width varies depending on *θ*. As a reference, the beam width at the inner end was set to *w*(6π) = 1.1 mm, and 3 types of spirals were fabricated, where the beam widths *w*(0) at the outermost end were 1, 3, and 4 times the reference width of *w*(6π). However, the change in the width from the spiral width *w*(0) to the spiral width *w*(6π) was a linear transition with respect to *θ*, i.e., $w\left(\theta \right)=\frac{\left(w\left(6\pi \right)-w\left(0\right)\right)}{6\pi }\theta +w\left(0\right)$. The maximum radius *r*_max_, the thickness *t*, and the beam width *w*(6π) of the spiral structure were 220 times larger than the design parameters of the previous THz chiral metamaterial [18,26]. Since the metal can be regarded as a perfect conductor at frequencies lower than the THz band, the structure size and the response frequency are basically inversely proportional. Therefore, these dimensions were designed to show a significant polarization effect at approximately 2.4 GHz, which is the frequency band of the antenna used in the following measurement. It should be noted that, if the spiral with a beam width ratio is *w*(0)/*w*(6π) = 4, the outermost beam of the spiral will touch the inner beam at the beam with *w*(6π) = 1.1 mm. To prevent this, we set *w*(6π) = 0.93 mm only for the spiral with the width ratio *w*(0)/*w*(6π) = 4. Moreover, if the beam width ratio *w*(0)/*w*(6π) was larger than 4, the spiral easily broke during deformation because the width was too small when the spiral width *w*(6π) was set to avoid contact; therefore, the beam width ratio *w*(0)*/w*(6π) ≤ 4 was verified.

To function as a metamaterial, the metamaterial’s structure should be arrayed. As shown in Figure 4, we arranged the spiral structures in a 2 × 2 configuration. In a chiral metamaterial, a *C*_4_ arrangement is usually adopted, where the neighboring metamaterials are rotated by 90° around the vertical axis such that the linear birefringence is eliminated from the evaluation of the optical activity. To verify the effect of the arrangement, *C*_1_ arranged metamaterials were also prepared. Metal was deposited only on the spiral structures to make them conductive and, thus, to cause them to function as a metamaterial. A vapor deposition system was used to deposit the metal film. The chamber of the evaporator was cylindrical with a diameter of 140 mm, so the device dimensions were set to 90 mm in both the length and width to fit in the chamber. At the 4 corners of the device, circular holes with a diameter of 13 mm were drilled to fix the device to the evaporator during vacuum deposition and to hold the device in an elongated spiral position.

### 2.4. Fabrication

The design of the fabricated spirals is shown in Figure 4a. A transparent acrylic sheet 90 mm in length and width and 0.5 mm in thickness was cut using a laser-processing machine to form spiral patterns, and aluminum was deposited on the spiral regions alone using a vacuum deposition system. Since the diameter of the spiral was 34.1 mm, i.e., 2*r*_max_, 4 spirals were cut in the same direction on a 90 mm square acrylic sheet. To deposit the metal only on the spiral region, the stencil mask shown in Figure 4b was fabricated using a laser machine in the same way as the spiral acrylic sheet. The acrylic sheet with the spiral with the stencil mask on was inserted into the vacuum deposition system, and aluminum film was deposited. The acrylic sheets were aligned using holes at the four corners. Several-hundred-nanometer-thick aluminum films were deposited on the spiral structures. Figure 5a shows the fabricated spiral acrylic sheet.

Next, the spiral structures processed on a flat surface were transformed into a three-dimensional spiral shape. For this purpose, another acrylic sheet of the same length and width as the sheet where the spiral structures were formed was prepared. Only the central part of the spiral was bonded to this acrylic plate with an adhesive. A three-dimensional spiral structure was formed by pulling up the sheet and fixing it with an attachment, as shown in Figure 5b. We made four attachments of different lengths to fix the height of the spiral to be 0, 16.5, 33, 49.5, and 66 mm. The maximum elongation of 66 mm is based on the preliminary calculation in [18], which shows that a practical level of polarization modulation capability can be obtained in terahertz. Therefore, the fabricated structure corresponded to the linear scaling up of the previously described chiral metamaterial.

### 2.5. Deformation Measurement

We measured the out-of-plane deformation profiles of the fabricated spiral structures by constructing a measurement setup using a guiding laser, as shown in Figure 6a. The visible laser was placed on the *z*-direction stage and it irradiated the spiral horizontally. The vertical displacement at the point where the light hits the spiral structure was measured from the micrometer level on the *z*-direction stage. Using this method, the height of the spiral structure at the azimuthal angle *θ* was measured with a *θ* interval of π/2. The measurement was performed for a spiral structure without an aluminum coating. Since the aluminum film is very thin, as small as a few hundred nanometers, its contribution to the stiffness of the structure is negligible. Under this condition, the amount of deformation of the spiral at each azimuth angle *θ* was measured by changing the deformation magnitude at the center of the spiral, i.e., the maximum deformation, to 33 mm, 49.5 mm, and 66 mm for a spiral with a constant beam width *w*(0)/*w*(6π) = 1, as shown in Figure 6b. Finally, the results were normalized by dividing the deformation by the maximum deformation in each structure. Here, the measurement results of the structure of the MEMS spiral metamaterial from [18] are superimposed. The theoretical value *δ*(*θ*) derived using Equation (4) is also plotted as a black line labelled as the “theoretical deformation”. The results show that the deformation profile of the spiral is similar regardless of the maximum deformation and follows the theory. Furthermore, the deformation profiles of the present structure are similar to those of the MEMS-sized structure, suggesting that the present large-scale structural analysis is an effective approach to deriving the optimal MEMS structure.

Next, we compared the deformation characteristics of the spirals when the beam width ratio *w*(0)/*w*(6π) was varied (Figure 6c). This is the normalized profile of 4 different spiral width ratio *w*(0)/*w*(6π) = 1, 2, 3, 4 structures deformed by 66 mm. The graph confirms that, as the spiral width ratio *w*(0)/*w*(6π) increases, the deformation of the spiral approaches a uniform deformation (represented by the dotted line in the graph), where the displacement is linear with respect to the azimuthal angle. However, although the spiral width ratio *w*(0)/*w*(6π) = 4 is the closest to uniform deformation, the approach to uniform deformation is saturated at the spiral width ratio *w*(0)/*w*(6π) = 2. It is also noted that, due to the end conditions, the deformation approaches uniform deformation at an azimuth angle smaller than π. By considering factors such as the end conditions in the model, it might be possible to make the deformation of the spiral closer to uniform deformation.

## 3. Results

### 3.1. Optical Measurement

To verify whether this deformation improved the chiral metamaterial’s response to circularly polarized light, we carried out transmission characteristic measurements in the radio frequency domain (Figure 7a). The response measurements were carried out in an anechoic chamber (TDK-EPC Corp., Tokyo, Japan) using a frequency band of approximately 2.4 GHz, which is commonly used for Wi-Fi and is readily available. The anechoic chamber is 7.4 m (L) × 2.2 m (W) × 3.7 m (H), the anechoic characteristic is smaller than −25 dB in the 2 GHz band, and the electromagnetic shielding characteristic is larger than 70 dB in the 1–18 GHz band. A vector network analyzer (E8364B, Agilent, Santa Clara, CA, USA) and antennas (FX-ANT-A1, CONTEC, Osaka, Japan) were used to transmit and receive the radio waves. Two hundred measurement points were set in the frequency range of 2–3 GHz. The transmitting and receiving antennas were connected to Port 1 and Port 2 of the vector network analyzer, respectively. The transmission and receiving antennas were placed on the upper and lower sides of the measurement enclosure, respectively, and a three-dimensional spiral structure was placed between them to measure the transmission characteristics, as shown in Figure 7b. The antenna positions were fixed in a Styrofoam box, of which the dielectric constant is close to a vacuum, and its influence on the measurement is negligible. The transmitting and receiving antennas were connected to Port 1 and Port 2 of the vector network analyzer, respectively. The distance between the transmitting antenna and the top of the device was fixed to 80 mm, and the distance between the transmitting and receiving antennas was fixed to 185 mm. The metamaterial device was approximately located at the center of space between the transmitting and receiving antennas. Under these conditions, transmission measurements were carried out for devices with different out-of-plane deformations of the 3D spiral structure.

In the transmission measurement, we focused on the transmission component of the cross-polarization, which is orthogonal to the incident electromagnetic wave polarization. As a peculiar property of the three-dimensional spiral structure, it is expected that a wave incident on the spiral structure with linear polarization will become elliptically polarized upon transmission due to the effect of the polarization rotation and the circular dichroism. It is supposed that the medium, i.e., the metamaterial, does not have birefringence for linearly polarized radio waves. Then, the chiral metamaterial’s effectiveness, the polarization rotation’s performance, and the circular dichroism can be evaluated by the transmittance of the polarization component in the orthogonal direction to the incident wave. The antennas used in this study, the FX-ANT-A1 transmitting and receiving antennas, can transmit and receive two orthogonal polarization components. In this experiment, the polarization parallel to the left–right direction is defined as V polarization, and the vertical polarization normal to the paper is defined as H polarization. The V polarization was transmitted from the transmitting antenna, and two types of polarization were measured at the receiving antenna. In the measurement, V polarization, which is the same polarization as the incident wave, and H polarization, which is the cross-polarization component, were measured. In this paper, we used the S-parameter value, which is the ratio of the amplitude of the incident wave voltage *V*_i_ leaving the point of Port 1 to the transmitted wave voltage *V*_t_ entering Port 2 via the receiving antenna and is defined in logarithmic notation as 20 log_10_ *V*_t_/*V*_i_.

### 3.2. Verification of Linear Birefringence in the C_1_ and C_4_ Structures

When a linearly polarized incident radio wave passes through a chiral metamaterial, the light’s polarization state becomes elliptical. In this study, this effect was evaluated by cross-polarization. If the structure has linear birefringence, the cross-polarization component is generated, and its amplitude is dependent on the rotation angle of the device around the normal vector to the device surface, which is the k vector direction. Thus, the circular polarization response cannot be evaluated appropriately. In this study, we prepared a device with a *C*_4_ configuration that does not have linear birefringence. Before evaluating the response characteristics of the device to circular polarization, we verified whether the *C*_4_ structure inhibits linear birefringence. For comparison, we prepared a *C*_1_ structure with all the spiral structures aligned in the same direction and compared the results. The device was rotated by 45° from 0° to 180° around the optical path axis, and the transmission spectra were compared. The spiral structures used for the comparison were spirals with a maximum deformation of 66 mm and a beam width ratio of *w*(0)/*w*(6π) = 3.

The results of the transmission measurements for the *C*_1_ structure and *C*_4_ are shown in Figure 8(a-*C*_1_) through (c-*C*_1_) and Figure 8(a-*C*_4_–c-*C*_4_), respectively. Figure 8(a-*C*_1_) and Figure 8(a-*C*_4_) are the results of the V polarization incident and V polarization transmitted measurements. The numbers 0~180° in the legend indicate the rotation angle around the normal direction of the device, as depicted in the upper right of Figure 8(a-*C*_1_). The legend “none” in the figure indicates transmission measurement results without any devices between them. The transmission graphs are convex upward because the antenna was for the 2.4 GHz band and has strong radiation near 2.4 GHz. Figure 8(b-*C*_1_) and Figure 8(b-*C*_4_) show the transmission measurement results of the V polarization incidence and H polarization transmission, i.e., cross-polarization transmission. Figure 8(c-*C*_1_) and Figure 8(c-*C*_4_) show the differential cross-polarization transmission, which is the difference from the cross-polarization results without the device (a condition of “none”).

First, we compared the results (a-*C*_1_) and (a-*C*_4_), which are the results for V polarization incidence and V polarization transmittance, and significant differences were not observed. The cross-polarization component is small, so the transmitted irradiation maintains its polarization state. The results in (b-*C*_1_), (b-*C*_4_), (c-*C*_1_), and (c-*C*_4_) show that the cross-polarization component is more significant in the frequency band above 2.3 GHz for both the *C*_1_ and *C*_4_ structures. Below 2.3 GHz, the cross-polarization is smaller than that of air, but the value is still small, indicating that there is practically no polarization rotation in this frequency band. The comparison between the *C*_1_ and *C*_4_ results shows that the *C*_1_ structure suffered from significant differences in the waveform depending on the rotation angle, while the *C*_4_ structure could eliminate the angle dependence over the entire measurement range. It was confirmed that the *C*_4_ structure removed the apparent polarization rotation effect caused by the linear birefringence.

### 3.3. Optical Response Change by Deformation

Next, changes in the transmission characteristics due to the degree of spiral deformation were measured. As in 3.2, linearly polarized light was incident on the spiral structure, and the transmission intensity of the radio wave was measured for two polarization conditions: V polarization incident and V polarization transmitted, and V polarization incident and H polarization transmitted, i.e., cross-polarization. In these measurements, *C*_4_ devices were employed. The transmission measurement results for the spiral structure with beam width ratios *w*(0)/*w*(6π) = 1 and 2 are shown in Figure 9(1-a) through (1-c) and (2-a) through (2-c), respectively. Figure 9(1-a,2-a) show the transmission intensities of V incidence and V transmittance. They show almost the same transmission intensity spectra even when the spiral deformation is changed, which is the same result as shown in Figure 8. On the other hand, when we focus on the transmitted component with the rotation of the polarization direction (Figure 9(1-b,2-b)), the transmitted intensity changes depending on the increasing deformation. To clarify this point, Figure 9(1-c,2-c) show the differential transmission spectra, which is the difference from the cross-polarization results without the device (“none” condition). In the case of *w*(0)/*w*(6π) = 1, the cross-polarization component increased for the deformation ranging from 16.5 to 49.5 mm over 2.5 GHz. On the other hand, in the case of *w*(0)/*w*(6π) = 2, the cross-polarization component increased for the deformation ranging from 16.5 to 49.5 mm over 2.3 GHz. The same spectral trends were observed for the devices with two beam width conditions. We next investigated the effect of the beam width ratio on circular dichroism.

### 3.4. Effect of the Beam Width

To investigate the effect of the beam width ratio, transmission spectra were extracted from data such as Figure 8 and Figure 9 for different beam width ratios *w*(0)/*w*(6π) =1, 2, 3, and 4 for deformations of 33 mm and 66 mm; they were then plotted on the same graph (Figure 10). The transmitted spectrum with *w*(0)/*w*(6π) = 2, 3, and 4 has a maximum intensity approximately 10 dB higher than the spectrum with *w*(0)/*w*(6π) = 1 at 2.3 GHz and above. On the other hand, there was no significant difference between spirals with beamwidth ratios of *w*(0)/*w*(6π) = 2, 3, and 4 (Figure 10(33 mm-b,66 mm-c)). This is consistent with the results of optical simulations, which show that increasing the beam width ratio causes the shape of the deformed spiral structure to approach an ideal shape, resulting in an increase in the circular dichroism (Figure 10(33 mm-c,66 mm-c)). Furthermore, as experimentally confirmed in Figure 6c, there is a clear difference in the three-dimensional shape of the spiral between *w*(0)*/w*(6π) = 1 and *w*(0)/*w*(6π) = 2, 3, and 4; this is consistent with the fact that there is no significant difference in shape between *w*(0)/*w*(6π) = 2, 3, and 4. The above experiments confirmed that, in millimeter-scale spiral structures, the properties of the optical response of the polarization effect improve as the spiral shape approaches homogeneity. We did not measure the polarization rotation angle, so it is not accurate, but we estimated the ellipticity simply from the ratio of the transmittance of V polarization incidence and V polarization transmittance, as well as V polarization incidence and H polarization transmittance in the data where the ellipticity is thought to be the highest, with *w*(0)/*w*(6π) = 4 and a deformation amount of 66 mm (Figure 10). At 2.5 GHz, the transmittance of V polarization incidence and V polarization transmittance is −18 dB, and the transmittance of V polarization incidence and H polarization transmittance is −30 dB, so the transmittance amplitude ratio is −12 dB, which translates to 0.25 when converted to a linear form. Taking the arctan of this ratio, there is a possibility that an ellipticity of about 14° is occurring. However, the above calculation is not accurate, and precise polarization measurements are necessary for an accurate evaluation. The above findings confirm that it is possible to evaluate the optical response of metamaterials with a three-dimensional spiral structure by experimentally verifying the structure conceived on the millimeter scale through the evaluation of radio wave responses in the GHz band. It is expected that the verification results obtained at the millimeter scale can be input back to the MEMS structure constructed at the micrometer scale.

## 4. Conclusions

In this study, we focused on the theoretical idea that spiral-shaped terahertz chiral metamaterials can improve the circular dichroism response with changes in their deformed shape; we conducted structural improvements of the deformed structure using macroscale devices. We constructed a structural mechanics model of the shape of the planar spiral structure when deformed and verified the effectiveness of the model by comparing it with the actual deformed structure. Based on the deformation behavior of the model, we attempted to make the root beam of the spiral thicker to achieve a more uniform deformation, which is expected to exhibit good circular dichroism. As a verification structure, we fabricated a large-scale spiral structure that operates in the GHz band by laser processing, which enabled speedy prototype verification. To evaluate the increase in circular dichroism, we irradiated the structure with linearly polarized radio waves and evaluated the cross-polarization component of the transmitted waves. As a result, we found that a more uniform deformation structure exhibits a higher cross-polarization component. This result suggests that uniform deformation structures can improve circular dichroism, which is consistent with the simulation results. Since terahertz metamaterials, such as GHz bands, can model metals as perfect conductors, a response in the THz band is expected to occur due to size shrinkage. Since large-scale devices enable speedy prototype verification, we plan to provide feedback on MEMS terahertz structures in the future.

## Figures and Tables

**Figure 1 micromachines-14-01156-f001:**
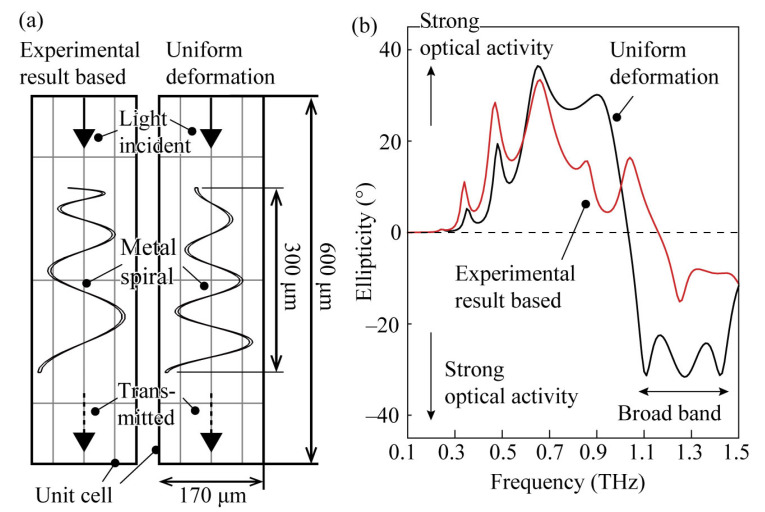
FEM calculation. (**a**) Simulation models with a spiral shape based on the experimental results and with a uniform deformation. (**b**) The ellipticity of two models. The shape based on the experimental result is based on a spiral shape measured experimentally by a laser microscope in the previous literature [18]. The shape obtained in the prior literature was proportionally enlarged in the height direction so that the amount of deformation from the base to the tip becomes 300 µm. On the other hand, the phenomenon referred to as uniform deformation means that, while the shape in the flat state before deformation is the same spiral, it deforms uniformly in the height direction relative to the azimuth angle of the spiral; ultimately, it is set to be 300 µm from the base to the tip.

**Figure 2 micromachines-14-01156-f002:**
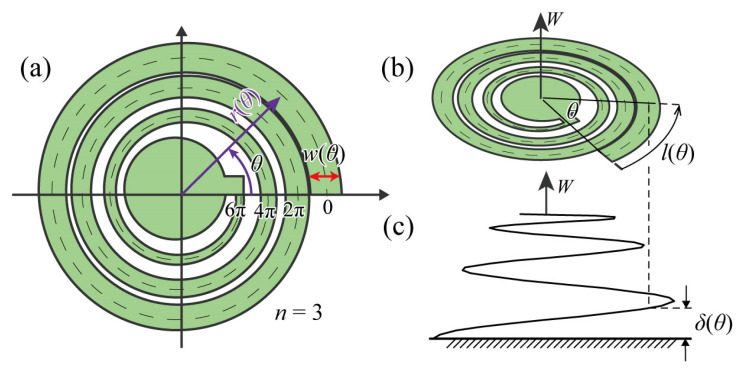
Model of the spiral structure, (**a**) a top view of the structure. The arrow indicates the direction of *θ* increase. (**b**) the structure from a bird’s-eye view point. The arrow beside *l*(*θ*) indicates the direction of length *l* increase, (**c**) the cross section of the deformed structure.

**Figure 3 micromachines-14-01156-f003:**
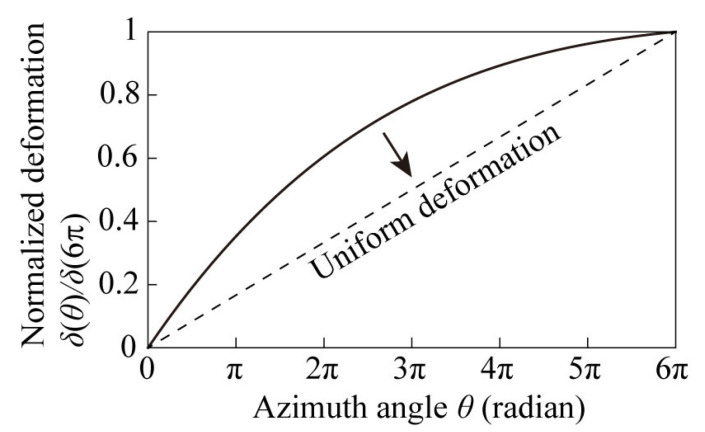
Calculated vertical deformation δ(θ).

**Figure 4 micromachines-14-01156-f004:**
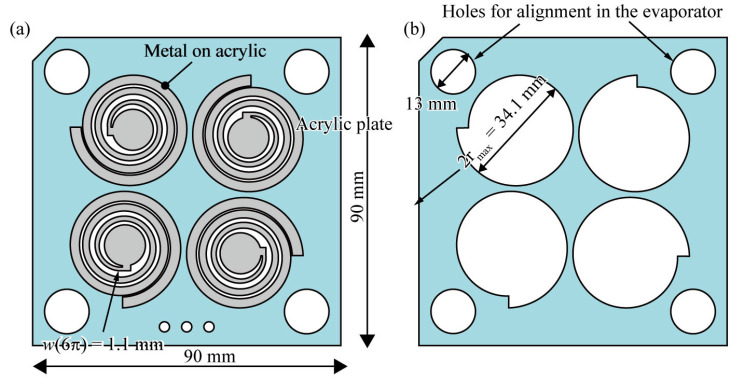
The device design; (**a**) the fabricated structure, and (**b**) a stencil for metal thermal evaporation.

**Figure 5 micromachines-14-01156-f005:**
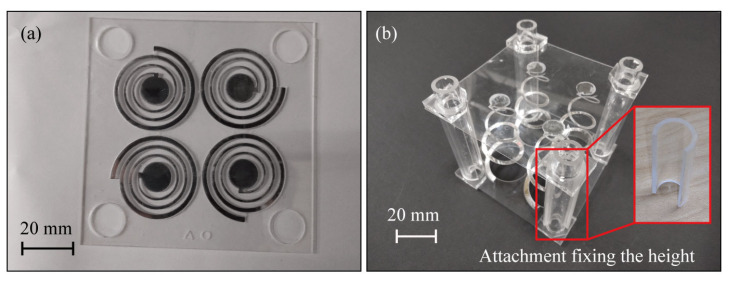
The fabricated device, (**a**) the fabricated acrylic sheet with planar spiral structures with metal deposition, and (**b**) the three-dimensional spirals deformed by pulling and fixing the structures.

**Figure 6 micromachines-14-01156-f006:**
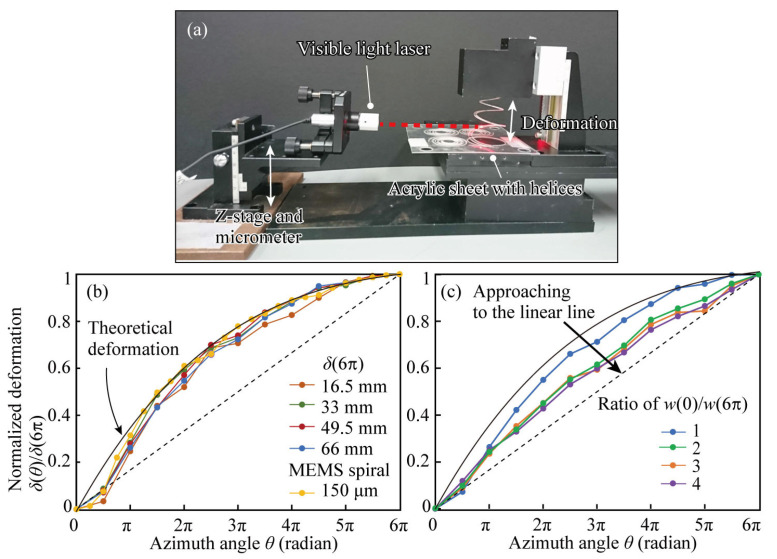
The deformation experiment of the fabricated spiral structures. (**a**) The experimental setup of the deformation measurement. The red dotted line indicates the laser path. (**b**) deformation profiles normalized by the maximum deformation heights and a comparison with the MEMS spiral deformation profile with respect to the azimuth angle. The MEMS spiral deformation was obtained the same MEMS spiral in [18]. (**c**) Normalized deformation profiles for different beam width ratios *w*(0)/*w*(6π).

**Figure 7 micromachines-14-01156-f007:**
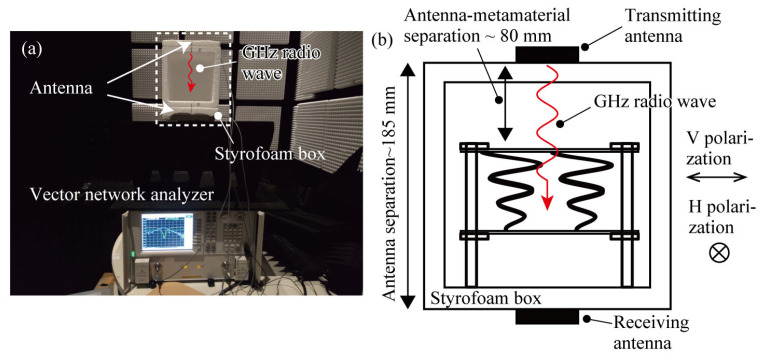
The transmittance experiment setup. (**a**) A photograph of the setup in the anechoic room, (**b**) a schematic of the antennas and the metamaterial arrangement embedded in the Styrofoam box.

**Figure 8 micromachines-14-01156-f008:**
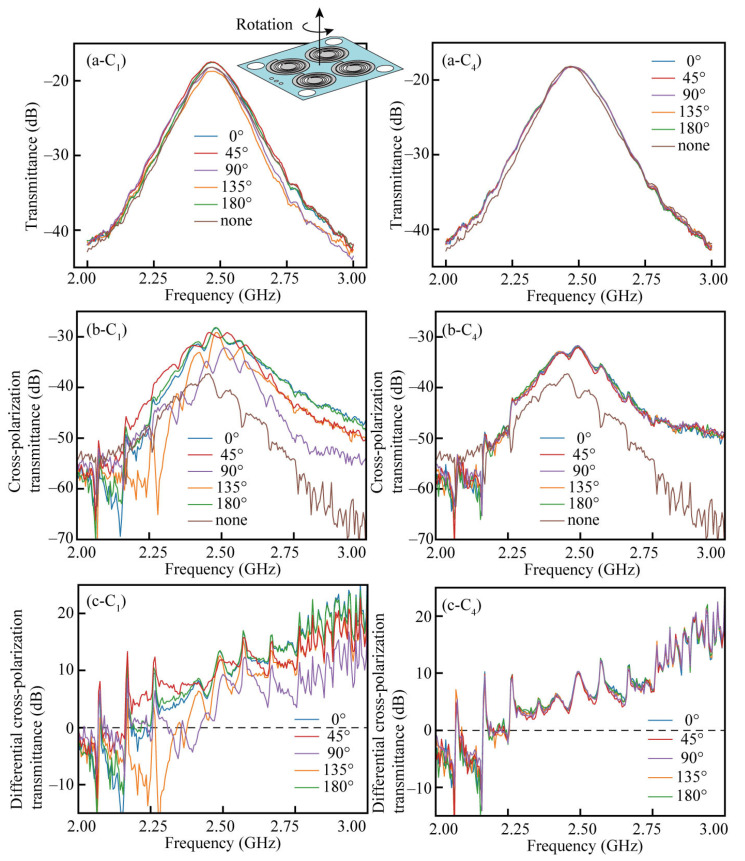
The transmittance measurement results. (**a-*C*_1_**,**a-*C*_4_**) The transmittance of the *C*_1_ and *C*_4_ devices, respectively, rotated around the normal axis to the device plane, (**b-*C*_1_**,**b-*C*_4_**) the cross-polarization transmittance of the *C*_1_ and *C*_4_ devices, respectively, (**c-*C*_1_**,**c-*C*_4_**) the differential cross-polarization transmission of the *C*_1_ and *C*_4_ devices, respectively.

**Figure 9 micromachines-14-01156-f009:**
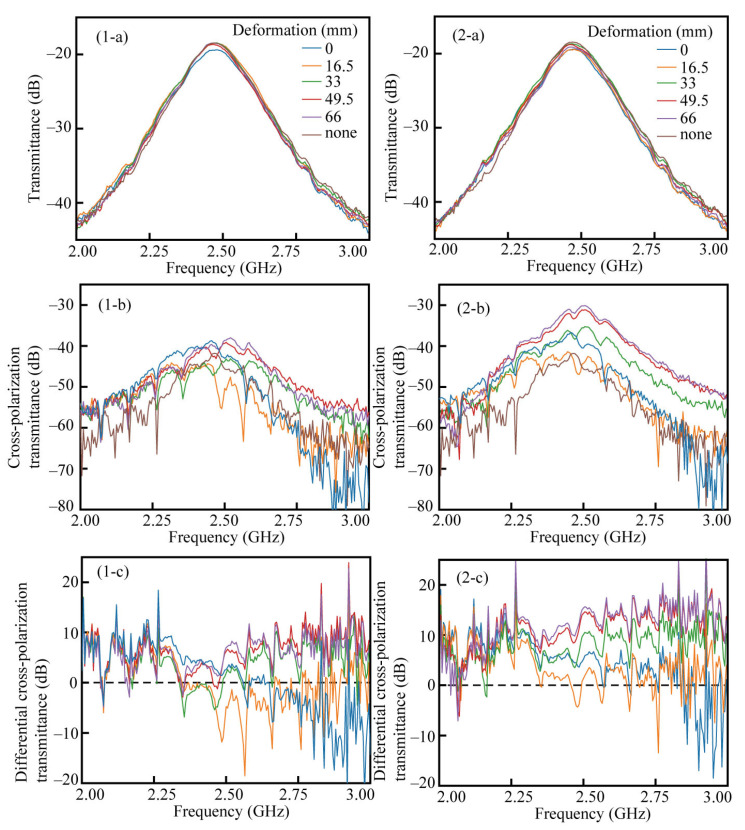
The transmittance measurement results: (**1**-**a**,**2**-**a**) the transmittance of the *C*_4_ devices with *w*(0)/*w*(6π) = 1 and 2, respectively, (**1**-**b**,**2**-**b**) the cross-polarization transmittance of the *C*_4_ devices with *w*(0)/*w*(6π) = 1 and 2, respectively, (**1**-**c**,**2**-**c**) the differential cross-polarization transmission of the *C*_4_ devices with *w*(0)/*w*(6π) = 1 and 2, respectively.

**Figure 10 micromachines-14-01156-f010:**
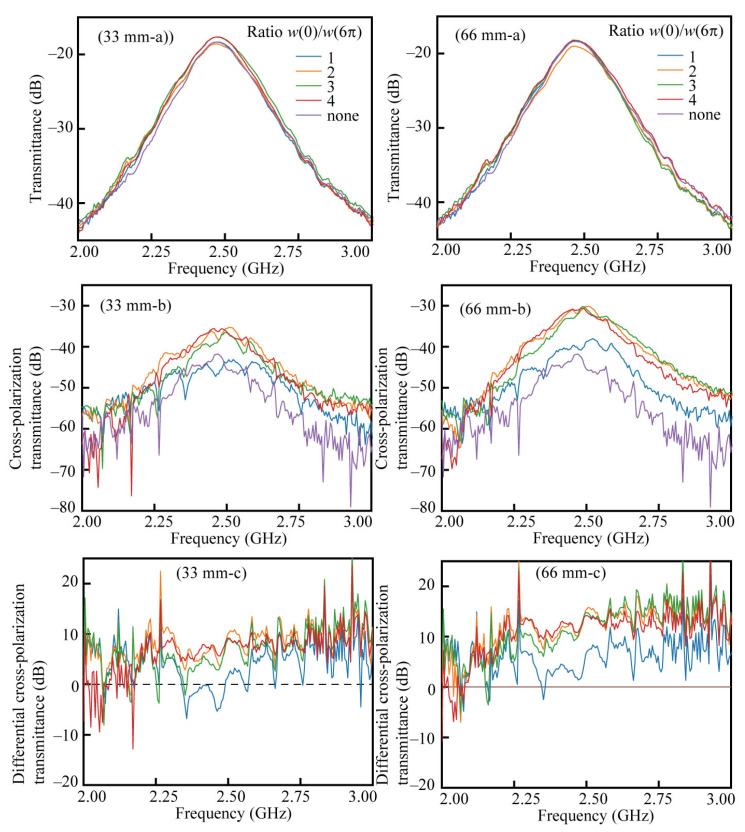
The transmittance measurement results, (**33 mm-a**,**66 mm-a**) the transmittance of the *C*_4_ devices with different beam ratios at the displacement values of 33 mm and 66 mm, respectively, (**33 mm-b**,**66 mm-b**) the cross-polarization transmittance of the *C*_4_ devices with different beam ratios at the displacement values of 33 mm and 66 mm, respectively, (**33 mm-c**,**66 mm-c**) the differential cross-polarization transmission of the *C*_4_ devices with different beam ratios at the displacement values of 33 mm and 66 mm, respectively.

## Data Availability

Data available on request.

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
