# Peer review of "Spiral Chiral Metamaterial Structure Shape for Optical Activity Improvements"

_micromachines, 2023, doi:10.3390/mi14061156_

Round 1

Reviewer 1 Report

This paper proposes uniform deformation of spiral metamaterials through shape optimization, and the resulting improvement in circular dichroism is reported numerically and experimentally. The article is well organized, and precise experiments were conducted. However, the completeness of the paper would be increased by adding further analysis and discussion. Detailed comments are as follows:

1. In Fig. 8, 9, and 10, experimental improvements are shown as differential cross-polarization transmittance. However, to prove that the proposed method is truly effective, it is necessary to discuss quantitatively how much angular variation in ellipticity can be taken, as shown in Fig. 1(b). This could be achieved by analyzing presented data with the method shown in Ref. [18].

2. In lines 335-336 (Page 11), the authors claim that "the transmitted intensity increases with an increasing deformation above 2.5GHz". However, the experimental results do not appear to be so simple. For example, In Fig. 9(2-b), around 2.5GHz, 0-16.5:decrease, 16.5-33-49.5: rapid increase, 49.5-66:saturation. This point should be explained.

Minor Issues

3. In line 355, Page 12, Figure 5(c) would be 6(c).

4. In Fig. 9(2-c), the negative signs of y-axis ticks are not shown.

5. In Fig. 1(b), The values of the y-axis of the graph would be incorrect (+-40 or +-45?).

(Optional) There are a few points where the polarization expression seems redundant." How about using the word "co-polarization" together with "cross-polarization"?

Reviewer 2 Report

In this paper, the authors have investigated a spiral structure that has the potential to obtain a large optical response. To study the shape of the planar spiral structure when deformed, the authors have built a structural mechanics model and confirmed its effectiveness. The authors have demonstrated in experiments that a more uniformly deformed structure had a higher cross-polarization component, indicating that uniform deformation structures can enhance circular dichroism. Overall, this paper is interesting and merits its publication in Micromachines. The reviewer recommends it to be accepted if the following comments could be properly addressed.

1. In the first figure of the paper, the authors investigated the spiral shape based on the experimental result and that with a uniform deformation. However, there’s no clear definition of these two types of shapes here, which may make it difficult for readers to understand the concepts being presented.

2. In this paper, Figure 6(b) shows that when the azimuth angle is small, the experimental measurements are closer to the curve of uniform deformation than the curve of theoretical deformation. Could the authors briefly explain this phenomenon?

3. It is presented in Figure 6(c) that when the ratio of w(0)/w(6pi) increases from 1 to 2, the movement of the curve is evident. However, as the ratio of w(0)/w(6pi) continues to increase from 2 to 4, the curves’ movement becomes negligible. In light of this, the reviewer questions whether the curves can approach the linear line.

The overall quality of the English language is good, although some minor editing may be needed.
